# SARS-CoV-2 seroprevalence among patients with severe mental illness: A cross-sectional study

Marie Reeberg Sass[1,2], Tobias Søgaard Juul[1,2], Robert Skov[3], Kasper Iversen[4], Lene Holm Harritshøj[5], Erik Sørensen[5], Sisse Rye Ostrowski[5], Ove Andersen[6], Claus Thorn Ekstrøm[7], Henrik Ullum[8], Jimmi Nielsen[2,9], Ida Hageman[2], Anders Fink-Jensen[1,2]*

1 Psychiatric Centre Copenhagen, Rigshospitalet, University of Copenhagen, Copenhagen, Denmark, 2 Mental Health Services, The Capital Region of Denmark, Copenhagen, Denmark, 3 Infection Disease Preparedness, Statens Serum Institut, Copenhagen, Denmark, 4 Department of Cardiology and Department of Emergency Medicine, Herlev and Gentofte Hospital, University of Copenhagen, Herlev, Denmark, 5 Department of Clinical Immunology, Rigshospitalet, University of Copenhagen, Copenhagen, Denmark, 6 Department of Emergency and Clinical Research Centre, Hvidovre Hospital, University of Copenhagen, Hvidovre, Denmark, 7 Department of Public Health, Section of Biostatistics, University of Copenhagen, Copenhagen, Denmark, 8 Statens Serum Institut, Copenhagen, Denmark, 9 Psychiatric Centre Glostrup, Mental Health Services, University of Copenhagen, Glostrup, Denmark

* anders.fink-jensen@regionh.dk

**Data Availability Statement:** All relevant data are within the manuscript. The full dataset containing personal information cannot be shared for confidentiality reasons.

## Abstract

Patients with severe mental illness (SMI) i.e. schizophrenia, schizoaffective disorder, and bipolar disorder are at increased risk of severe outcomes if infected with coronavirus disease 2019 (COVID-19). Whether patients with SMI are at increased risk of COVID-19 is, however, sparsely investigated. This important issue must be addressed as the current pandemic could have the potential to increase the existing gap in lifetime mortality between this group of patients and the background population. The objective of this study was to determine whether a diagnosis of schizophrenia, schizoaffective disorder, or bipolar disorder is associated with an increased risk of COVID-19. A cross-sectional study was performed between January 18th and February 25th, 2021. Of 7071 eligible patients with schizophrenia, schizoaffective disorder, or bipolar disorder, 1355 patients from seven psychiatric centres in the Capital Region of Denmark were screened for severe acute respiratory syndrome coronavirus 2 (SARS-CoV-2) antibodies. A total of 1258 unvaccinated patients were included in the analysis. The mean age was 40.5 years (SD 14.6), 54.3% were female. Fifty-nine of the 1258 participants had a positive SARS-CoV-2 antibody test, corresponding to a adjusted seroprevalence of 4.96% (95% CI 3.87–6.35). No significant difference in SARS-CoV-2-risk was found between female and male participants (RR = 1.32; 95% CI 0.79–2.20; p = .290). No significant differences in seroprevalences between schizophrenia and bipolar disease were found (RR = 1.12; 95% CI 0.67–1.87; p = .667). Seroprevalence among 6088 unvaccinated blood donors from the same region and period was 12.24% (95% CI 11.41–13.11). SARS-CoV-2 seroprevalence among included patients with SMI was significantly lower than among blood donors (RR = 0.41; 95% CI 0.31–0.52; p < .001). Differences in seroprevalences remained significant when adjusting for gender and age, except for those aged 60

**Funding:** The study was funded by Mental Health Services, the Capital Region of Denmark. The funders played no role in the study design, data collection and analysis, decision to publish, or preparation of the manuscript.

**Competing interests:** The authors have declared that no competing interests exist.

years or above. The study is registered at ClinicalTrails.gov (NCT04775407). https://clinicaltrials.gov/ct2/show/NCT04775407?term=NCT04775407&draw=2&rank=1.

## Introduction

The pandemic of coronavirus disease 2019 (COVID-19) caused by severe acute respiratory syndrome coronavirus 2 (SARS-CoV-2) has caused health problems worldwide [1]. More than 250 million cases have been confirmed globally [2]. As the pandemic has evolved, it has become increasingly evident that individuals are disproportionately affected by the disease, e.g. patients with comorbid diabetes, obesity, and cardiovascular diseases have been reported to correlate with a worse outcome [3–8]. The same somatic conditions are overrepresented in patients with a severe mental illness (SMI) [9,10]. In addition, patients suffering from SMI are at increased risk for infectious diseases, have lower hospitalization rates, and have limitations in access to healthcare [11–13]. Thus, patients with SMI are possibly at increased risk of severe outcomes of COVID-19. These concerns have been confirmed in several studies [14–20]. However, whether patients with SMI are more likely to be infected with SARS-CoV-2 is not clear [14–16,21,22].

There is a need for studies to elucidate if this patient group is at increased risk of contracting the virus and whether the COVID-19 pandemic increases the existing health inequalities between this vulnerable patient group and the general population. The aim of the present study was to determine the seroprevalence of SARS-CoV-2 antibodies in patients with a diagnosis of schizophrenia, schizoaffective disorder, or bipolar affective disorder receiving in-patient or out-patient care via mental health services in the Capital Region of Denmark and to compare these data with the seroprevalence among Danish blood donors as a proxy for the general population in the Capital Region of Denmark. Additionally, we aimed to examine possible risk factors that might be associated with SARS-CoV-2 infection.

## Methods

### Study overview

This cross-sectional study was conducted at seven psychiatric centres in the Capital Region of Denmark. The Scientific-Ethical Committee of the Capital Region of Denmark (H-20037171) and the Danish Data Protection Authorities (P-2020-1037) in the Capital Region of Denmark approved the study. The study was carried out in accordance with the Declaration of Helsinki 1964 and with national laws and Regulations for Clinical Research. The study is registered at ClinicalTrails.gov: NCT04775407.

### Participants

Eligible patients were adults aged 18 or above, diagnosed with SMI i.e. schizophrenia, schizoaffective disorder, or bipolar affective disorder according to the criteria of the International Classification of Diseases, World Health Organization (WHO), and treated in the Capital Region of Denmark. To secure sufficient inclusion of patients, all seven psychiatric centres in The Mental Health Services, Capital Region of Denmark, provided the following information (from EPIC/Medical Record System); name, social security number, contact information, diagnoses, and name on the psychiatric department. Patients who were found too vulnerable for participation by their treating psychiatrists were excluded.

## Study design

Between January 18[th] and February 19[th], 2021 all eligible patients were invited to serological screening for SARS-CoV-2 antibodies. All patients who fulfilled the inclusion criteria received an invitation and written study information through the Danish governmental, personal, secure email system (E-Boks) or by letter. All patients were then contacted via phone and oral study information was provided. Contact via phone was attempted up to three times. Patients whose phone number was not available through EPIC/Medical Record System, received another E-Boks message with contact information to study investigators. All patients were offered up to 48 hours to consider participation.

The serological screening took place from February 1[st], through February 25[th], 2021, at 20 different locations, associated with seven psychiatric centres throughout the Capital Region of Denmark. Patients were free to choose test day and location. On testing day, patients who wanted to participate signed written consent. All patients were informed that they could withdraw from the study at any time without providing a reason and without any consequences for their current or future psychiatric treatment. Patients who did not manage to show up for the scheduled blood test were offered a new appointment.

A maximum of 6 mL blood was collected per patient, and serum was tested for total SARS-CoV-2 immunoglobulins by use of WANTAI SARS-CoV-2 antibody ELISA [23]. Analyses were performed at Statens Serum Institut in Copenhagen, Denmark, and carried out according to the manufacturer's recommendations (Beijing Wantai Biological Pharmacy Enterprise Co., Ltd.). The manufacturer reported a sensitivity of 94.5% and a specificity of 100% [23]. All samples were analysed within 8 weeks and destroyed immediately afterward. Test results were categorized as positive if they showed the presence of SARS-CoV-2 immunoglobulins.

Results from included patients with SMI were compared with data from Danish blood donors in the Capital Region of Denmark obtained from the same period. Data extracted from the database of Danish blood donors, included results of SARS-CoV-2 antibody tests and demographic data, i.e. age and gender. These data were used as a proxy for the general population in the Capital Region of Denmark. Inconclusive test results and results from study participants and blood donors who had received at least one dose of any vaccine against SARS-CoV-2 were excluded from the statistical analysis.

## Statistical analysis

Seroprevalences were presented as percentages. Demographic data were presented descriptively, e.g. mean, standard deviation (SD), and percentages and compared (seropositive vs. seronegative and participants vs. non-participants) using unpaired t-tests and Fisher's exact tests respectively. P-values for Fisher's exact tests were two-sided. Comparisons between included patients and blood donors were explored through relative risks (RRs). RRs were calculated as the probability of having a positive test for SARS-CoV-2 antibodies among included patients compared with the probability among blood donors. We calculated RRs for gender and age groups to rule out the risk of these factors being confounders. Further, RRs were calculated to explore associations between SARS-CoV-2 serology and certain risk factors, such as gender, diagnosis, geographical area, and age <30 years. RRs and seroprevalences were calculated using the "twoby2" function in the R-package "Epi". The 95% confidence intervals (CIs) of the probabilities were computed using the normal approximation to the log-odds as implemented in the "Epi" package. All seroprevalences were adjusted for test sensitivity and specificity, using the Rogan and Gladen [24] method through the R package "epiR". Results were presented with 95% CIs. All p-values below 0.05 were considered statistically significant. Statistical analyses were calculated using R (http://www.R-project.org/).

## Results

### Trial population and baseline characteristics

Out of a total population of 7310 patients, 238 patients were found too vulnerable for participation. Of the remaining, 1355 patients (19.2%) signed the consent. Of these, 41 (3.0%) of consenters had received at least one dose of a COVID-19 vaccine. Blood samples from another 56 (4.1%) patients were not included in the analysis due to various reasons (Fig 1). A total of 1258 patients (17.2% of the total study population) were included in the final analysis, of which 54.3% were female. The mean age was 40.5 years (SD 14.6). Seropositive patients were significantly younger than seronegative patients (mean age (SD): 35.7 (13.5) vs. 40.8 (14.6); p = .006).

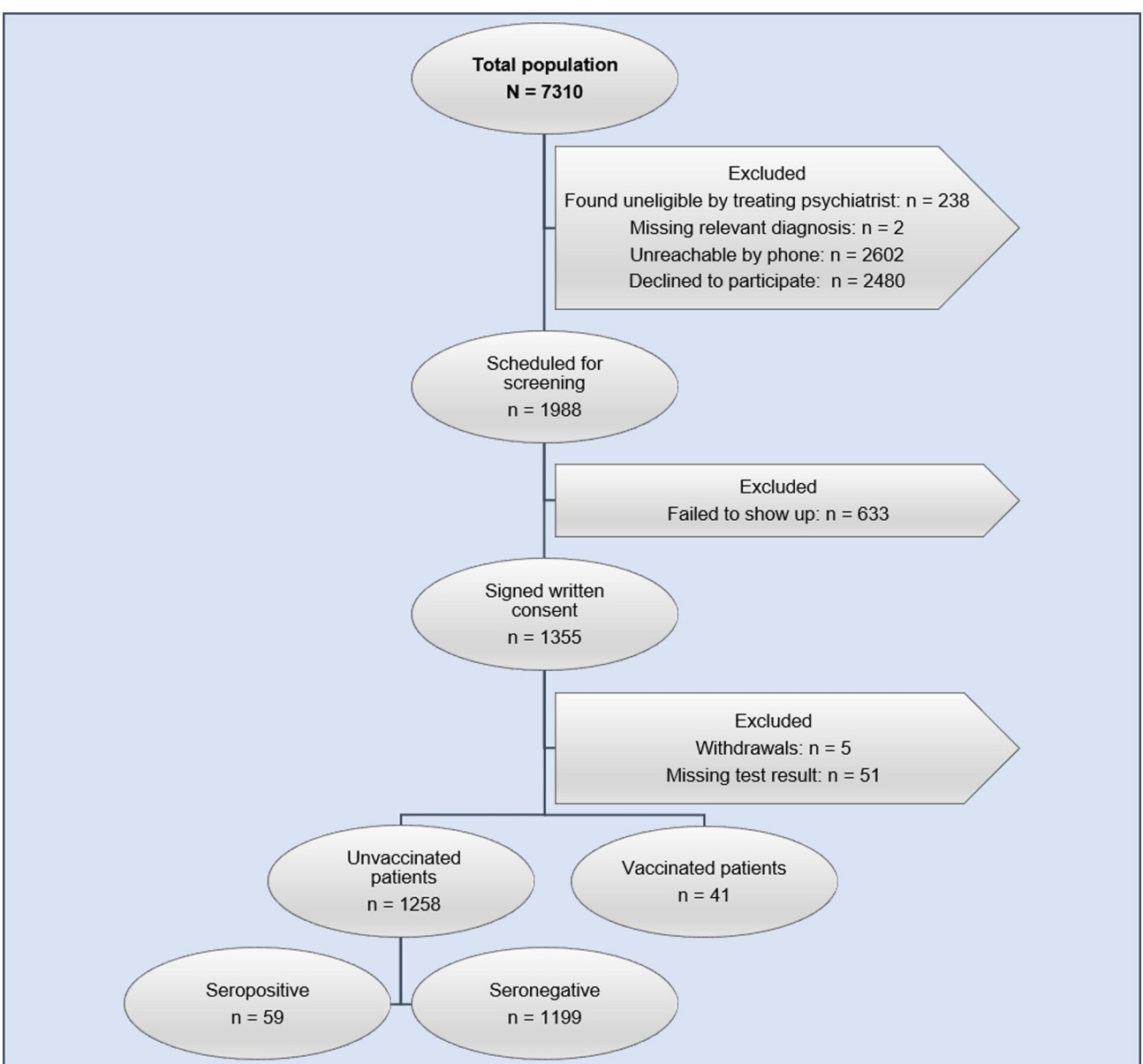

**Fig 1. Flowchart of patients with severe mental illness.** All unvaccinated patients with severe mental illness who fulfilled all criteria to be enrolled in the study, and with a positive or negative SARS-CoV-2 antibody test result, were included in the statistical analysis.

**Table 1. Baseline characteristics of study population according to SARS-CoV-2 serology[a].**

| Characteristic | Total (N = 1258) | Positive (n = 59) | Negative (n = 1199) | P value[b] |
|---|---|---|---|---|
| **Age, mean (SD)** | 40.5 (14.6) | 35.7 (13.5) | 40.8 (14.6) | .006 |
| **Gender, n (%)** | | | | .349 |
| Female | 683 (54.3) | 36 (61.0) | 647 (54.0) | |
| Male | 575 (45.7) | 23 (39.0) | 552 (46.0) | |
| **Diagnosis, n (%)** | | | | .316 |
| Schizophrenia | 727 (57.8) | 37 (62.7) | 690 (57.5) | |
| Schizoaffective disorder | 47 (3.7) | 0 | 47 (3.9) | |
| Bipolar disorder | 484 (38.5) | 22 (37.3) | 462 (38.5) | |
| **Geographical location, n (%)** | | | | .002 |
| Copenhagen municipality[c] | 711 (56.5) | 45 (76.3) | 666 (55.5) | .10 |
| Suburban municipalities[d] | 547 (43.5) | 14 (23.7) | 533 (44.5) | |

[a]All patients with severe mental illness who underwent SARS-CoV-2 antibody testing and were included in the statistical analysis.

[b]Comparisons between seropositive and seronegative participants calculated using Fisher's exact test and unpaired t-test. Proportions may not be equal to 100% due to rounding error.

[c]Including psychiatric centres of Copenhagen and Amager.

[d]Including psychiatric centres of Ballerup, Bornholm, Glostrup, Nordsjælland, and Sct. Hans.

Most of the included patients (57.8%) were diagnosed with schizophrenia. Baseline characteristics of the included patients are presented in Table 1. Participants were significantly younger (mean age (SD): 40.5 (14.6) vs. 45.0 (16.4); p < .001) and more participants were females (54.3% vs. 46.8%; p < .001) and diagnosed with bipolar disorder (38.5% vs. 24.5%; p < .001) compared with SMI patients who did not participate in the study (Table 2).

## SARS-CoV-2-seroprevalence

Fifty-nine of 1258 included patients had a positive SARS-CoV-2 antibody test, corresponding to a adjusted seroprevalence of 4.96 (95% CI 3.87–6.35). No significant difference was found

**Table 2. Baseline characteristics of total study population by study participation[a].**

| Characteristic | Total (N = 7310) | Participants (n = 1258) | Non-participants (n = 6052) | P value[b] |
|---|---|---|---|---|
| **Age, mean (SD)** | 44.2 (16.2) | 40.5 (14.6) | 45.0 (16.4) | < .001 |
| **Gender, n (%)** | | | | < .001 |
| Female | 3516 (48.1) | 683 (54.3) | 2833 (46.8) | |
| Male | 3794 (51.9) | 575 (45.7) | 3219 (53.2) | |
| **Diagnosis, n (%)** | | | | < .001 |
| Schizophrenia | 5025 (68.7) | 727 (57.8) | 4298 (71.0) | |
| Schizoaffective disorder | 316 (4.3) | 47 (3.7) | 269 (4.4) | |
| Bipolar disorder | 1969 (26.9) | 484 (38.5) | 1485 (24.5) | |
| **Geographical location, n (%)** | | | | < .001 |
| Copenhagen municipality[c] | 3604 (49.3) | 711 (56.5) | 2893 (47.8) | |
| Suburban municipalities[d] | 3706 (50.7) | 547 (43.5) | 3159 (52.2) | |

[a]All patients treated for either schizophrenia, schizoaffective disorder, or bipolar disorder in The Mental Health Services, Capital Region of Denmark.

[b]Comparisons between seropositive and seronegative participants calculated using Fisher's exact test and unpaired t-test. Proportions may not be equal to 100% due to rounding error.

[c]Including psychiatric centres of Copenhagen and Amager.

[d]Including psychiatric centres of Ballerup, Bornholm, Glostrup, Nordsjælland, and Sct. Hans.

**Table 3. Associations between SARS-CoV-2 antibodies and certain risk factors among study participants.**

| Risk factor | Seroprevalences, % (95% CI)[a] | | Relative Risk (95% CI) | P-value |
|---|---|---|---|---|
| | Risk factor group | Control group | | |
| Female[b] | 5.58 (4.05–7.63) | 4.23 (2.84–6.28) | 1.32 (0.79–2.20) | .290 |
| Schizophrenia[c] | 5.39 (3.93–7.34) | 4.81 (3.20–7.18) | 1.12 (0.67–1.87) | .667 |
| Copenhagen municipality[b] Calibri (Brødtekst) | 6.70 (5.04–8.85) | 2.71 (1.62–4.50) | 2.47 (1.37–4.46) | .002 |
| Age <30[b] | 7.08 (4.76–10.40) | 4.17 (3.02–5.72) | 1.70 (1.02–2.82) | .042 |

[a]Seroprevalences were adjusted for test sensitivity and specificity.

[b]Control group = Remaining participants.

[c]Control group = Participants with bipolar disease.

between the participating females and males (RR = 1.32; 95% CI 0.79–2.20; p = .290). The diagnose of schizophrenia were not associated with a higher risk of COVID-19 infection, compared with bipolar disease (RR = 1.12; 95% CI 0.67–1.87; p = .667). No cases (95% CI 0.00–8.00) of SARS-COV-2 antibodies were detected among patients with schizoaffective disorder. Patients from districts located in Copenhagen municipality were at higher risk of COVID-19 compared with patients from suburban psychiatric centres (RR = 2.47; 95% CI 1.37–4.46; p = .002). Also, participants younger than 30 years were at significantly higher risk of being seropositive compared with those aged 30 or above (RR = 1.70; 95% CI 1.02–2.82; p = .042) (Table 3).

In the same period, 6088 (50.0% female, mean age 40.0 years (SD 13.7)) unvaccinated blood donors from the Capital Region of Denmark were tested for SARS-CoV-2 antibodies. A total of 704 blood donors were seropositive, corresponding to a adjusted seroprevalence of 12.24 (95% CI 11.41–13.11). Seroprevalence among participants in the present study was significantly lower than among blood donors (RR = 0.41; 95% CI 0.31–0.52; p < .001) (Fig 2).

When comparing gender, significant differences between included patients and blood donors remained (female: RR = 0.43; 95% CI 0.31–0.60; p < .001, male: RR = 0.37; 95% CI 0.24–0.56; p < .001) (Table 4). As Table 4 reveals, the seroprevalence was significantly lower for patients with SMI across all age groups except for patients aged 60 years or above (<30 years: RR = 0.41, 95% CI 0.27–0.62; p < .001, 30–39 years: RR = 0.49; 95% CI 0.30–0.79; p = .003, 40–49 years: RR = 0.32, 95% CI 0.16–0.64; p = .001, 50–59 years: RR = 0.38; 95% CI 0.18–0.81; p = .010, ≥60 years: RR = 0.51; 95% CI 0.18–1.42; p = .194).

## Discussion

The present study investigated the seroprevalence of SARS-CoV-2 antibodies in patients with SMI compared with the prevalence among blood donors in the same region and period. We found that the risk of SARS-CoV-2 seropositivity was significantly lower among patients with SMI than among blood donors. The risk of COVID-19 remained significantly lower among the patients when comparing across gender and age groups apart from for those aged 60 or above. The seroprevalence was significantly higher in SMI patients younger than 30 and patients from psychiatric centres located in Copenhagen municipality with no difference between diagnoses or gender. This is in congruence with the patterns of SARS-CoV-2 seroprevalence among the general Danish population [25].

The significantly lower seroprevalence of SARS-CoV-2 antibodies in patients with SMI may be a consequence of a lower number of social contacts for patients with psychotic disorders and patients with increased anxiety and depression symptoms related to the pandemic [26–28]. While isolation due to lock-down has imposed profound changes on many people's

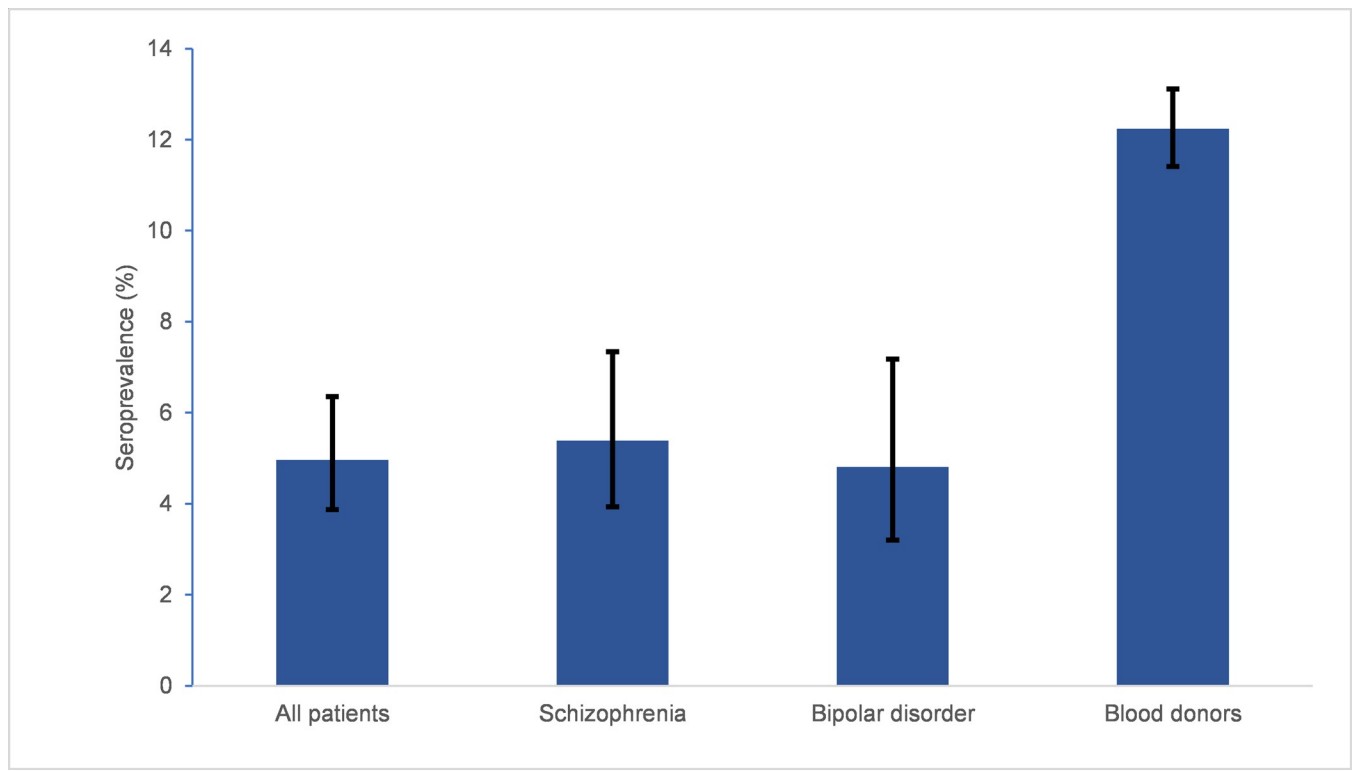

**Fig 2. Seroprevalences for included patients with severe mental illness compared with blood donors of the Capital Region of Denmark.** Seroprevalences adjusted for test sensitivity and specificity with 95% confidence intervals among all included patients with severe mental illness (n = 1258), according to ICD-10 diagnosis (schizophrenia n = 725, bipolar disorder n = 484) and blood donors (n = 6088).

private and working life, the changes for people with SMI may have been less profound. In accordance with our findings, Van der Meer et al. have reported, that patients with psychiatric disorders were more frequently SARS-CoV-2-negative than people undergoing testing without such conditions. The population in this study, however, was small and not fully representative [22]. On the contrary, other studies have reported an association between SMI and increased risk of contracting COVID-19 [14,16,21].

In our study, we included seroprevalence from unvaccinated blood donors. Limitations using blood donors as a proxy for the general population must be considered [29]. A study has

**Table 4. Seroprevalences and RRs for the total group and according to gender and age group for study participants and blood donors.**

| Characteristic | Seroprevalence, % (95% CI)[a] | | Relative Risk (95% CI) | P-value |
|---|---|---|---|---|
| | **SMI patients** | **Blood donors** | | |
| Total | 4.96 (3.87–6.35) | 12.24 (11.41–13.11) | 0.41 (0.31–0.52) | < .001 |
| Female | 5.58 (4.05–7.63) | 13.02 (11.84–14.31) | 0.43 (0.31–0.60) | < .001 |
| Male | 4.23 (2.84–6.28) | 11.45 (10.33–12.67) | 0.37 (0.24–0.56) | < .001 |
| Age <30 | 7.08 (4.76–10.40) | 17.15 (15.47–18.97) | 0.41 (0.27–0.62) | < .001 |
| Age: 30–39 | 5.92 (3.73–9.28) | 12.20 (10.41–14.25) | 0.49 (0.30–0.79) | .003 |
| Age: 40–49 | 3.25 (1.65–6.28) | 10.30 (8.65–12.22) | 0.32 (0.16–0.64) | .001 |
| Age: 50–59 | 3.65 (1.78–7.35) | 9.57 (7.99–11.44) | 0.38 (0.18–0.81) | .010 |
| Age: ≥60 | 2.92 (1.14–7.28) | 5.72 (4.08–7.97) | 0.51 (0.18–1.42) | .194 |

[a]Seroprevalences were adjusted for test sensitivity and specificity.

proposed that seropositivity might be lower in blood donors than in the general Danish population, as they represent a healthy group, screened for behavioral risks [30,31]. However, it has been estimated that 30% of COVID-19 transmission in Denmark occurs at workplaces [32]. One could argue that blood donors due to their better health conditions, could constitute a larger proportion of the labor market and thus at higher risk of SARS-CoV-2.

In a recent National Danish surveillance study, a 2 percentage points lower seroprevalence (10.69% (95% CI 9.74–11.73)), than for our blood donors samples, was found among 3919 participants from the Capital Region of Denmark. The surveillance study was conducted over 3 weeks following the termination of our own study [25]. This suggests that the seroprevalence among danish blood donors could be slightly higher than in the general Danish population.

The majority of previously published studies report an increased risk of SARS-CoV-2 infection among patients with SMI [14,16,21], an association strongest for schizophrenia [14,21]. In our study, we found a significantly low seroprevalence independently of mental disorder. These contradictory results can be attributed to various reasons. To the best of our knowledge, all previous studies were designed as retrospective cohorts with results based on database registered PCR tests which induce a risk of selection bias; In some countries, infection symptoms have been a prerequisite for COVID-19-testing, why asymptomatic cases would not be captured. In a meta-analysis, Buitrago-Garcia et al. [33] estimated that approximately 20% of people with COVID-19 infection remain asymptomatic. Additionally, it is well recognised that patients with SMI find it difficult to report physical health needs [34,35]. Furthermore, there may be COVID-19-testing fees that deter people from being tested as well as the use of other non-national test forms (e.g. pop-up sites with antigen rapid tests) which may not be registered in the national electronic health records. Due to our study design with the use of serological antibody screening, selection bias as a consequence of various test strategies has been avoided. A variety of other factors are also likely to contribute to conflicting results regarding the risk of COVID-19 among patients with SMI. Findings might be influenced by the characteristics of the Health Care System. Appraising health care information can be challenging for patients with SMI [36]. Muruganandam et al. reported that knowledge about COVID-19 infection is reduced in this patient population compared with the general population, with two-thirds of the SMI patients not having adequate knowledge about precautionary measures [37]. All patients, regardless of the type of SMI, receiving in-patient or out-patient care via one of the psychiatric centres in the Capital Region of Denmark have a personal health care professional who during lockdown has been able to provide COVID-19 related information during the COVID-19 lockdown. The close interaction between patients in our study population and health care professionals might have played an important role in optimizing the patients understanding of the pandemic and thereby minimizing their risk of infection. Socioeconomic differences (e.g. education level and housing conditions) for patients with SMI between countries must also be considered as part of the explanation [38]. As an example, homelessness in Denmark is associated with a higher risk of SARS-CoV-2 infection [39]. In Denmark however, all patients with SMI are offered social assistance and housing. Homelessness as a risk factor for patients with SMI may play a larger role in countries with a less comprehensive social welfare system.

Out of the total number of eligible patients in the study population, 1355 patients chose to participate. Serological samples from 1258 unvaccinated patients were included in the final analysis. The participation rate in our study was relatively high considering the vulnerable patient population (17.2% of the total patient population). Still, it must be considered if study participants are representative of patients with SMI in general. There was a significantly larger proportion of females and patients with bipolar disorder among participants compared with non-participants. This might have biased our results, as previous studies have found an

increased risk of SARS-CoV-2 infections among patients with schizophrenia compared with other psychiatric conditions [14]. However, the fact that we found no differences in seroprevalence between diagnoses and gender makes this unlikely. We excluded serological results from patients who had received at least one dose of any COVID-19 vaccine. At this stage in Denmark, only patients expected to be at high risk of a severe outcome of COVID-19 were offered vaccination [40]. This may have biased our results if this vulnerable group of patients, e.g. psychotic or manic patients, were at higher risk of infection. Nevertheless, if vaccinated patients were included in the analysis, the seroprevalence (7.90%) for our study participants remains significantly lower than for unvaccinated blood donors (RR = 0.65; 95% CI 0.53–0.79; p < .001). The number of COVID-19-related deaths among Danish patients with schizophrenia or bipolar affective disorder is relatively low [20] and therefore unlikely to be a reason for the difference in seroprevalence.

Several studies have addressed that patients with SMI, are at increased risk of severe outcomes of COVID-19 [14–20] and common comorbid risk factors among patients with psychiatric disorders such as chronic obstructive pulmonary disease, obesity, and cardiovascular disease, are associated with severe COVID-19 outcomes [9,10]. A recently published study reported a 2 fold higher risk of severe outcomes for patients with schizophrenia and bipolar disorder, respectively across all regions in Denmark [20]. Consequently, the current pandemic has the potential to increase the already existing gap in lifetime mortality between this group of patients and the background population. Our results suggest that patients with SMI might be less exposed to COVID-19, but the fact that these patients are at higher risk of severe COVID-19 outcomes emphasizes the importance of continued focus on this vulnerable patient group during the pandemic.

## Limitations and strengths

This study has several strengths. Our study design allowed us to compare seroprevalences, for two demographically similar populations in the exact same period. Although our study population comprises a mentally vulnerable group of patients, our participation rate is comparable with those of the nationwide SARS-CoV-2 prevalence studies of the general Danish population [25,41]. To our knowledge, this is the first study systematically screening patients diagnosed with schizophrenia, schizoaffective disorder, and bipolar disorder for SARS-CoV-2 antibodies. By testing for SARS-CoV-2 antibodies, we secured that individuals having had asymptomatic infections with SARS-CoV-2 were also included. Finally, all blood samples were analysed using ELISA, which has high sensitivity and specificity and therefore a low risk of false-positive or false-negative test results. The present study also has limitations that must be considered. Patients who previously had been tested positive by PCR or antigen rapid tests might have declined to participate thereby underestimating our results. To minimize this bias, however, the importance of study participation regardless of former infection was emphasized to invited patients when recruited to the study. Furthermore, it is worth mentioning, that the same bias exists in the previously mentioned National Danish Surveillance Study. Due to the cross-sectional design, it was not possible to determine when the participants were infected. Also, recently infected patients may not have been detected, as antibodies are detectable around one week after contamination [42]. Out of the total study population, 238 patients were found too vulnerable to participate by their treating psychiatrist. Furthermore, patients on closed wards or under involuntary commitment were not allowed to participate. Both groups might be at higher risk of SARS-CoV-2 infection. Study participants were, nevertheless, younger and more comprised of patients from the psychiatric centres in Copenhagen municipality compared with non–participants in the study population—both factors which were associated with higher seroprevalence in our study participants.

In conclusion our study documents a significantly lower seroprevalence of SARS-CoV-2 antibodies among patients with SMI compared with the seroprevalence of SARS-CoV-2 in blood donors. Further studies should be carried out to elucidate the different impacts of COVID-19 in patients with SMI between countries to secure national health care strategies.

## Acknowledgments

We want to express our gratitude to Alexandra Stephanie Zamorski, Tanja Pedersen, and Cyril Martel who helped with planning and logistics. Additionally, we want to thank the patients and the seven psychiatric centers in the Capital Region of Denmark for participating in the study. TestCenter Denmark is thanked for the SARS-CoV-2 antibody analysis of blood samples.

## Author Contributions

**Conceptualization:** Marie Reeberg Sass, Tobias Søgaard Juul, Robert Skov, Kasper Iversen, Ove Andersen, Claus Thorn Ekstrøm, Henrik Ullum, Jimmi Nielsen, Ida Hageman, Anders Fink-Jensen.

**Data curation:** Marie Reeberg Sass, Tobias Søgaard Juul, Robert Skov, Kasper Iversen, Lene Holm Harritshøj, Erik Sørensen, Sisse Rye Ostrowski, Ove Andersen, Claus Thorn Ekstrøm, Henrik Ullum, Jimmi Nielsen, Ida Hageman, Anders Fink-Jensen.

**Formal analysis:** Marie Reeberg Sass, Tobias Søgaard Juul, Lene Holm Harritshøj, Erik Sørensen, Sisse Rye Ostrowski, Claus Thorn Ekstrøm, Anders Fink-Jensen.

**Funding acquisition:** Anders Fink-Jensen.

**Investigation:** Marie Reeberg Sass, Tobias Søgaard Juul, Lene Holm Harritshøj, Erik Sørensen, Sisse Rye Ostrowski, Jimmi Nielsen, Anders Fink-Jensen.

**Methodology:** Marie Reeberg Sass, Tobias Søgaard Juul, Robert Skov, Kasper Iversen, Lene Holm Harritshøj, Erik Sørensen, Sisse Rye Ostrowski, Ove Andersen, Claus Thorn Ekstrøm, Henrik Ullum, Jimmi Nielsen, Ida Hageman, Anders Fink-Jensen.

**Project administration:** Marie Reeberg Sass, Tobias Søgaard Juul, Anders Fink-Jensen.

**Resources:** Robert Skov, Henrik Ullum, Jimmi Nielsen, Ida Hageman, Anders Fink-Jensen.

**Software:** Jimmi Nielsen.

**Supervision:** Anders Fink-Jensen.

**Validation:** Anders Fink-Jensen.

**Visualization:** Marie Reeberg Sass, Tobias Søgaard Juul, Anders Fink-Jensen.

**Writing – original draft:** Marie Reeberg Sass, Tobias Søgaard Juul, Anders Fink-Jensen.

**Writing – review & editing:** Robert Skov, Kasper Iversen, Lene Holm Harritshøj, Erik Sørensen, Sisse Rye Ostrowski, Ove Andersen, Claus Thorn Ekstrøm, Henrik Ullum, Jimmi Nielsen, Ida Hageman.

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
