## [Decision Letter · Decision Letter 0]

12 Oct 2021

PONE-D-21-26611SARS-CoV-2 seroprevalence among patients with severe mental illness: a cross-sectional studyPLOS ONE

Dear Dr. Sass,

Thank you for submitting your manuscript to PLOS ONE. After careful consideration, we feel that it has merit but does not fully meet PLOS ONE’s publication criteria as it currently stands. Therefore, we invite you to submit a revised version of the manuscript that addresses all the points raised below during the review process.

We look forward to receiving your revised manuscript.

Kind regards,

Ray Borrow, Ph.D., FRCPath

Academic Editor

PLOS ONE

Journal Requirements:

“This study was funded by Mental Health Services, the Capital Region of Denmark.”

We note that you have provided funding information within the Acknowledgements Section. Please note that funding information should not appear in the Acknowledgments section or other areas of your manuscript. We will only publish funding information present in the Funding Statement section of the online submission form.

“The study was funded by Mental Health Services, the Capital Region of Denmark. The funders played no role in the study design, data collection and analysis, decision to publish, or preparation of the manuscript.”

Reviewers' comments:

Reviewer's Responses to Questions

**Comments to the Author**

1. Is the manuscript technically sound, and do the data support the conclusions?

Reviewer #1: Partly

Reviewer #2: Yes

Reviewer #3: Partly

Reviewer #4: Yes

2. Has the statistical analysis been performed appropriately and rigorously? 

Reviewer #1: No

Reviewer #2: I Don't Know

Reviewer #3: Yes

Reviewer #4: No

3. Have the authors made all data underlying the findings in their manuscript fully available?

Reviewer #1: Yes

Reviewer #2: Yes

Reviewer #3: Yes

Reviewer #4: Yes

4. Is the manuscript presented in an intelligible fashion and written in standard English?

Reviewer #1: Yes

Reviewer #2: Yes

Reviewer #3: Yes

Reviewer #4: Yes

5. Review Comments to the Author

Reviewer #1: The manuscript entitled ‘The SARS- CoV-2 seroprevalence among patients with severe mental illness: a cross-sectional study’ with the aim to determine whether a diagnosis of schizophrenia, schizoaffective disorder, or bipolar disorder is associated with an increased risk of COVID-19.

This is quite an interesting study but the manuscript can be further improved.

Abstract

Page 3 Line 47, the comma to indicate 95% CIs could be omitted e.g 95% CI, 0.79-2.20 can be presented as 95% CI 0.79-2.20 or 95% CI: 0.79-2.20.

Page 3 Line 49, comparison of gender to be highlighted.

Methods

The type of SMI included in the study to be clearly stated.

Statistical analyses

Line 121, the word respectively to be added after tests.

Line 121, unpaired t-tests and Pearson’s chi-square tests to be written as unpaired t-test and Pearson’s chi-square test. Likewise with Line 154-155, 161-162.

Line 124, the sentence ‘to adjust for gender and age, we calculated RRs for gender and age groups’ requires revision. The word adjust not clear.

Line 129, the word adjust to be replaced with another word. Likewise with Line 187.

Results

Table 1, the empty column to be removed.

Line 143-144, the sentence requires revision.

Line 151, the title for Table 1 is too short.

Table 1, decimal point for p value to be standardized. N to be used for overall sample size (Total). No.(%) to be replaced with n(%).

Line 166-175, Line 181-189, data/results to be presented in table form. The analysis whether crude or adjusted to be clearly stated. Multivariable statistical test could be explored.

The cited reference to be spaced out from the previous word throughout the manuscript e.g. COVID-19(14,16,21). [ ] to be used instead of ( ).

List of references to conform with PLoS ONE format.

Reviewer #2: Line 99 and Figure 2 – While Blood donors were used as a comparison, is there an estimate of the average population rate of COVID positivity in Denmark in February 2021? And if so, is this different? Blood donors may not be representative of the general population and this should be addressed in the text. The authors should discuss more why the patients in this study had a lower than population rate of positivity but, as noted in the Discussion, probably a higher rate of hospitalization due to COVID-19.

In general the results suggest that mental health patients are significantly less likely to have contracted COVID-19 is a surprising finding and contrary to most other studies. Clearly the study is limited by the patients who chose to be screened and may have not agreed to screening because they knew they had had COVID-19. This complicates the interpretation of the results unless there is another reason for the difference. Enhanced discussion is needed

Reviewer #3: The manuscript is well written and studies an underserved population (severe mentally ill). The focus on COVID in this population is highly timely. Investigation of such a population is challenging for many reasons and the authors have done an excellent job describing their approach and success.

A major strength of the study is reliance on serology for estimates of prevalence, which is not subject to many bias and other potential confounding factors related to medical care, patient perception, isolation, and testing of the mentally ill.

The study findings of a low prevalence of COVID (~4%) in the mentally ill population is remarkable and worth recognition. The factors that contribute to the low prevalence are important to recognize. This reviewer feels the low prevalence rate alone should be a major focus, rather than comparison to another group.

The major weakness of the study is the attempt to compare their study population to a reference group. While of interest, the rate of blood bank donors is a poor comparison group. The rate of similarly institutionalized individuals would make a much better comparison group, of which, admittedly there are few. One suggestion might be prisoners.

Another analytic approach might be to compare the mentally ill patients to multiple other groups that may have been published, such as local COVID rates in Denmark among citizens, or specialized groups (health care workers). The limitation of the comparison group is redundantly addressed in the discussion and the limitations sections.

Reviewer #4: This study is original as it examines the association between severe mental disorders and risk of COVID-19. The author uses a cross-sectional study on psychiatric centers with blood donors as control. The study is of interest even if the results are not significant.

I have some questions for this manuscript including the following (organized by manuscript section).

Methods - Statistical analysis

Due to the small number of expected subjects (<5) in some categories, and the highly unbalanced marginal sums of the actual data set, Pearson’s chi-square test cannot be applied. It is better to use Fisher’s exact test.

Results

All seroprevalences in the text are crude prevalences and not corrected prevalences as suggested by Rogan et al. 1978 and indicated in the method paragraph. So give the corrected prevalences even if there are not significantly different from the crude prevalence. Do the same in the abstract.

I don’t find the same prevalences than authors for male and female:

Crude prevalence = 4.69% for female (32/683) instead of 5.27% in the text

Corrected prevalence= 4.97% for female (0.0469+1-1)/(0.945+1-1) according to Rogan et al. 1978)),

Crude prevalence = 4.70% for male (27/575) instead of 4.00% in the text

Corrected prevalence= 4.97% for male

Even if there are no cases of SARS-COV-2 antibodies among patients with schizoaffective disorder, it is possible to calculate the 95% CI.

For mental health diagnosis, it will be better to calculate prevalence for each diagnosis separately, and relative risk as compared to schizophrenia which is the largest group, and not as compared to remaining included patients with SMI which include people with different prevalences.

For the different districts of Copenhagen, I would separate central part and Amager from the other districts which have higher corrected prevalences of respectively 6.67 and 6.78 while others have corrected prevalences of 2.74.

Even if it seems evident, gives the reference in the sentence 174-175 about age. It could be another group than ≥ 30 years.

RR between participants and blood donors is 0.41 instead of 0.40 with approximation (0.405).

Table 1: there is an extra empty column. Thank you to delete it. I will be interested to have age by group and not as a continued variable, the mean age being included in the text.

Discussion

The paragraph lines 218- must be just after lines 211.

I will be interesting to know if there are some difference in the literature according severe mental health diagnosis and to discuss it and not only according sex differences.

It would also be interesting to discuss the difference between neighbourhoods. Is it differences in socio-economic level, in population characteristics?

Problems at the end of line 259 for reference 20.

6. PLOS authors have the option to publish the peer review history of their article (what does this mean?). If published, this will include your full peer review and any attached files.

Reviewer #1: No

Reviewer #2: No

Reviewer #3: No

Reviewer #4: No

---

## [Author Response · Author response to Decision Letter 0]

12 Nov 2021

We thank the Reviewers for taking the time to carefully read our submission to PLOS ONE and appreciate the positive and constructive comments. In the following, please find the specific comments from Reviewers and the corresponding responses from the authors. Thank you.

Reviewer #1: The manuscript entitled ‘The SARS- CoV-2 seroprevalence among patients with severe mental illness: a cross-sectional study’ with the aim to determine whether a diagnosis of schizophrenia, schizoaffective disorder, or bipolar disorder is associated with an increased risk of COVID-19.

This is quite an interesting study but the manuscript can be further improved.

We do appreciate the constructive and positive comments by Reviewer 1 and appreciate hers/his time reviewing our manuscript.

Abstract

Page 3 Line 47, the comma to indicate 95% CIs could be omitted e.g 95% CI, 0.79-2.20 can be presented as 95% CI 0.79-2.20 or 95% CI: 0.79-2.20.

Author response: Revised accordingly. Thank you for your comment.

Page 3 Line 49, comparison of gender to be highlighted.

Authors response: Please see lines 44-45: “No significant difference in SARS-CoV-2-risk was found between female and male participants (RR=1.32; 95% CI 0.79-2.20; p=.29).” We hope that you will find this adequate.

Methods

The type of SMI included in the study to be clearly stated.

Authors response: Please see lines 32 and 82. We hope that you will find this adequate.

Statistical analyses

Line 121, the word respectively to be added after tests.

Author response: Thank you. Revised accordingly.

Line 121, unpaired t-tests and Pearson’s chi-square tests to be written as unpaired t-test and Pearson’s chi-square test. Likewise with Line 154-155, 161-162.

Author response: Thank you. Revised accordingly.

Line 124, the sentence ‘to adjust for gender and age, we calculated RRs for gender and age groups’ requires revision. The word adjust not clear.

Author response: The sentence has been rewritten: “We calculated RRs for gender and age groups to rule out the risk of these factors being confounders.”

Line 129, the word adjust to be replaced with another word. Likewise with Line 187.

Author response: Replaced with “corrected”.

Results

Table 1, the empty column to be removed.

Author response: You are completely right. Thank you for your comment, the empty column has been removed. 

Line 143-144, the sentence requires revision.

Author response: Thank you. The sentence has now been rephrased: “Participants were significantly younger (mean age (SD): 40.5 (14.6) vs. 45.0 (16.4); p<.001) and more participants were females (54.3% vs. 46.8%; p<.001) and diagnosed with bipolar disorder (38.5% vs. 24.5%; p<.001) (Table 2) compared with SMI patients who did not participate in the study.”

Line 151, the title for Table 1 is too short.

Author response: Good point. The title has been changed from: “Baseline characteristics” to “Baseline characteristics of patients according to SARS-CoV-2 serology”.

Table 1, decimal point for p value to be standardized. N to be used for overall sample size (Total). No.(%) to be replaced with n(%).

Author response: Corrected. P-values � .01 are presented with 2 decimals, p-values between .001 and .01 are presented with 3 decimals and p-values below .001 are presented as p<.001. 

Line 166-175, Line 181-189, data/results to be presented in table form. The analysis whether crude or adjusted to be clearly stated. Multivariable statistical test could be explored.

Author response: Results are now presented in table forms with corrected seroprevalences. Unfortunately, it was not possible to do multivariable tests, as data from blood donors were sent categorically in tables and not from each individual. 

The cited reference to be spaced out from the previous word throughout the manuscript e.g. COVID-19(14,16,21). [ ] to be used instead of ( ). List of references to conform with PLoS ONE format.

Author response: We apologize and have carefully revised the manuscript, and the references do now confirm with PLOS ONE format.

Reviewer #2: Line 99 and Figure 2 – While Blood donors were used as a comparison, is there an estimate of the average population rate of COVID positivity in Denmark in February 2021? And if so, is this different? Blood donors may not be representative of the general population and this should be addressed in the text.

Author response: Thank you for your time and effort. We agree that this comparison would be very relevant. Unfortunately, data for the average population rate of COVID positivity in Denmark in February 2021 does not exist. Instead, we have added to the manuscript a comparison of our results with data from a National Danish Surveillance study conducted over 3 weeks right after termination of our own study. Please see lines 220-224. The reviewer is completely right in his/her comment about the limitations of using blood donors as a reference group and we believe to have discussed this thoroughly. Please see lines 214-219.

The authors should discuss more why the patients in this study had a lower than population rate of positivity but, as noted in the Discussion, probably a higher rate of hospitalization due to COVID-19.

Author response: Somatic conditions associated with a worse outcome of COVID-19 (e.g. hospitalization) are overrepresented in patients with SMI. Reasons for this are enhanced in the discussion lines 391-399. Also, please see see introduction lines 58-61. We believe that our contradictory results with a low seroprevalence among patients with SMI can be attributed to various reasons; Isolation (lines 205-209), selection bias in testing strategies (lines 236-237), characteristics of the Health Care System (lines 242-248), Socioeconomic differences (lines 248-252). Thank you.

In general the results suggest that mental health patients are significantly less likely to have contracted COVID-19 is a surprising finding and contrary to most other studies. Clearly the study is limited by the patients who chose to be screened and may have not agreed to screening because they knew they had had COVID-19. This complicates the interpretation of the results unless there is another reason for the difference. Enhanced discussion is needed.

Author response: This is a good point and a possible bias that also exists in the National Surveillance studies. We have enhanced our discussion please see lines 290-293. Thank you.

Reviewer #3: The manuscript is well written and studies an underserved population (severe mentally ill). The focus on COVID in this population is highly timely. Investigation of such a population is challenging for many reasons and the authors have done an excellent job describing their approach and success.

A major strength of the study is reliance on serology for estimates of prevalence, which is not subject to many bias and other potential confounding factors related to medical care, patient perception, isolation, and testing of the mentally ill.

The study findings of a low prevalence of COVID (~4%) in the mentally ill population is remarkable and worth recognition. The factors that contribute to the low prevalence are important to recognize. This reviewer feels the low prevalence rate alone should be a major focus, rather than comparison to another group.

The major weakness of the study is the attempt to compare their study population to a reference group. While of interest, the rate of blood bank donors is a poor comparison group. The rate of similarly institutionalized individuals would make a much better comparison group, of which, admittedly there are few. One suggestion might be prisoners. Another analytic approach might be to compare the mentally ill patients to multiple other groups that may have been published, such as local COVID rates in Denmark among citizens, or specialized groups (health care workers). The limitation of the comparison group is redundantly addressed in the discussion and the limitations sections.

Author response: We thank you for your positive and constructive comments. We do agree that our finding of a low prevalence in the population of SMI patients is important and the key point in our publication. The perspective of our approach, using a comparison group, is to highlight this result. Our rationale with a comparison of the low prevalence to another group is exactly to make the common reader see, how surprisingly low the prevalence of COVID-19 among patients with SMI is. We do agree that it would have been preferable to have a comparison group without limitations. To this end, we have now included a reference from a National Danish Surveillance study (please see lines 220-224) conducted over 3 weeks in the extension of our study. Comparing our results to the data from participants in the National Danish surveillance study still results in a significantly lower seroprevalence in our patient group. Patients participating in this study are not institutionalized as prisoners but part of the general community. We have included a reference to a study that found that homelessness in Denmark is associated with a higher risk of COVID-19 (please see lines 249-252). In Denmark with a comprehensive social welfare system, however, this group would not be a preferable group for comparison, due to socioeconomic differences (SMI patients in Denmark have normally their own apartment. All people have free access to the hospital system). The data from blood donors were collected in the exact same period and region as blood samples were drawn from patients participating in the study. We, therefore, of the possible reference groups, do believe to have chosen the best group for comparison to the general population to highlight our result of a low seroprevalence among patients with SMI.

Reviewer #4: This study is original as it examines the association between severe mental disorders and risk of COVID-19. The author uses a cross-sectional study on psychiatric centers with blood donors as control. The study is of interest even if the results are not significant. 

I have some questions for this manuscript including the following (organized by manuscript section).

We do appreciate the constructive and positive comments by Reviewer 4 and appreciate hers/his time reviewing our manuscript.

Methods - Statistical analysis

Due to the small number of expected subjects (<5) in some categories, and the highly unbalanced marginal sums of the actual data set, Pearson’s chi-square test cannot be applied. It is better to use Fisher’s exact test.

Author response: Fisher’s exact test could not be made with too many groups or observations. To make Fisher’s exact test possible, the psychiatric centres are now separated in those located in Copenhagen municipality and those located elsewhere. Thank you for the comment.

Results

All seroprevalences in the text are crude prevalences and not corrected prevalences as suggested by Rogan et al. 1978 and indicated in the method paragraph. So give the corrected prevalences even if there are not significantly different from the crude prevalence. Do the same in the abstract.

Author response: All seroprevalences are now presented as corrected prevalences. Thank you.

I don’t find the same prevalences than authors for male and female:

Crude prevalence = 4.69% for female (32/683) instead of 5.27% in the text

Corrected prevalence= 4.97% for female (0.0469+1-1)/(0.945+1-1) according to Rogan et al. 1978)),

Crude prevalence = 4.70% for male (27/575) instead of 4.00% in the text

Corrected prevalence= 4.97% for male

Author response: Thank you for this observation. You are completely right. The seroprevalences for males and females were correct but the numbers in Table 1 were not. They are now corrected. 

Even if there are no cases of SARS-COV-2 antibodies among patients with schizoaffective disorder, it is possible to calculate the 95% CI.

Author response: The 95% CI of 0.00-8.00 is now added to the manuscript. 

For mental health diagnosis, it will be better to calculate prevalence for each diagnosis separately, and relative risk as compared to schizophrenia which is the largest group, and not as compared to remaining included patients with SMI which include people with different prevalences.

Author response: Results are now presented according to each diagnosis, while relative risks are calculated between schizophrenia patients and patients with bipolar disease.

For the different districts of Copenhagen, I would separate central part and Amager from the other districts which have higher corrected prevalences of respectively 6.67 and 6.78 while others have corrected prevalences of 2.74.

Author response: Thank you for your observation. Districts are now separated in psychiatric centres located in Copenhagen municipality (Copenhagen and Amager), and those located in the suburbs (Ballerup, Bornholm, Glostrup, Nordsjælland, and Sct. Hans).

Even if it seems evident, gives the reference in the sentence 174-175 about age. It could be another group than ≥ 30 years.

Author response: Good point. We have now added: ”compared with those aged 30 or above”. 

RR between participants and blood donors is 0.41 instead of 0.40 with approximation (0.405).

Author response: Rounding error is corrected. Thank you.

Table 1: there is an extra empty column. Thank you to delete it. I will be interested to have age by group and not as a continued variable, the mean age being included in the text.

Author response: Thank you for your comment, the empty column is now deleted. Age by group in table 1 and table 2 would make it impossible to do comparison tests (Fisher’s exact tests), as there would be too many groups/observations. 

Discussion

The paragraph lines 218- must be just after lines 211.

Author response: Thank you. This has now been corrected 

It will be interesting to know if there are some difference in the literature according severe mental health diagnosis and to discuss it and not only according sex differences. It would also be interesting to discuss the difference between neighbourhoods. Is it differences in socio-economic level, in population characteristics?

Author response: Thank you for the interesting perspective. We have enhanced our discussion according to severe mental health diagnosis (please see lines 225-226 + 258-260). We have added that in Denmark all patients with severe mental illness, regardless of the type of mental diagnosis, are offered social assistance and housing. This could explain why we see no difference in seroprevalence between the groups of patients with SMI (i.e schizophrenia, -schizoaffective disorder, and -bipolar disorder) in our population. The seroprevalence was significantly higher among patients from Psychiatric Centres located in Copenhagen municipality which is the area in Denmark with the highest population density. This is in congruence with the patterns of SARS-CoV-2 seroprevalence among the general Danish population due to data from a National Danish Surveillance Study (please see lines 220-224). The offer of social services to patients with SMI does not differ between neighborhoods in Denmark. Unfortunately, however, we do not have any exact knowledge or numbers on socio-economic differences between the groups in our population. Hopefully, we can explore this in future studies. 

Problems at the end of line 259 for reference 20.

Author response: Thank you. This has now been corrected.

6. PLOS authors have the option to publish the peer review history of their article (what does this mean?). If published, this will include your full peer review and any attached files.

Do you want your identity to be public for this peer review? For information about this choice, including consent withdrawal, please see our Privacy Policy.

Reviewer #1: No

Reviewer #2: No

Reviewer #3: No

Reviewer #4: No

---

## [Decision Letter · Decision Letter 1]

12 Jan 2022

PONE-D-21-26611R1SARS-CoV-2 seroprevalence among patients with severe mental illness: a cross-sectional studyPLOS ONE

Dear Dr. Sass,

Thank you for submitting your manuscript to PLOS ONE. After careful consideration, we feel that it has merit but does not fully meet PLOS ONE’s publication criteria as it currently stands. Therefore, we invite you to submit a revised version of the manuscript that addresses the last remaining points raised below during the review process.

We look forward to receiving your revised manuscript.

Kind regards,

Ray Borrow, Ph.D., FRCPath

Academic Editor

PLOS ONE

Journal Requirements:

Reviewers' comments:

Reviewer's Responses to Questions

**Comments to the Author**

1. If the authors have adequately addressed your comments raised in a previous round of review and you feel that this manuscript is now acceptable for publication, you may indicate that here to bypass the “Comments to the Author” section, enter your conflict of interest statement in the “Confidential to Editor” section, and submit your "Accept" recommendation.

Reviewer #1: All comments have been addressed

Reviewer #2: All comments have been addressed

Reviewer #3: All comments have been addressed

Reviewer #4: (No Response)

2. Is the manuscript technically sound, and do the data support the conclusions?

Reviewer #1: (No Response)

Reviewer #2: Yes

Reviewer #3: Yes

Reviewer #4: Yes

3. Has the statistical analysis been performed appropriately and rigorously? 

Reviewer #1: (No Response)

Reviewer #2: Yes

Reviewer #3: Yes

Reviewer #4: No

4. Have the authors made all data underlying the findings in their manuscript fully available?

Reviewer #1: (No Response)

Reviewer #2: Yes

Reviewer #3: Yes

Reviewer #4: Yes

5. Is the manuscript presented in an intelligible fashion and written in standard English?

Reviewer #1: (No Response)

Reviewer #2: Yes

Reviewer #3: Yes

Reviewer #4: Yes

6. Review Comments to the Author

Reviewer #1: The word adjusted could still be used as alternative to the word corrected.

For Table 1 & 2, chi-squares test where applicable could still be utilized provided the assumptions of the statistical test are fulfilled/met. Please check the assumptions and also recheck the p values. If Fisher's Exact test is used, one sided or two sided p value to be stated.

Reviewer #2: Revisions were adequate Revisions were adequate Revisions were adequate Revisions were adequate Revisions were adequate

Reviewer #3: (No Response)

Reviewer #4: I thank the authors for their responses to comments. Nevertheless, I have some other comments.

Introduction

If you want to use the acronym SMI, it is best to associate it with severe mental illness as in the abstract. Otherwise use SMD for severe mental disorders.

Methods - Statistical analysis

Due to the small number of expected subjects (<5) in some categories, and the highly unbalanced marginal

sums of the actual data set, Pearson’s chi-square test cannot be applied. It is better to use Fisher’s exact test.

Author response: Fisher’s exact test could not be made with too many groups or observations.

Reviewer comment : The extension of Fisher's exact test to the case where the two variables have any finite number of modalities, but greater than two, was first performed by G. H. Freeman and J.H. Halton in 1951. This test is sometimes referred as the Freeman-Halton or Fisher-Freeman-Halton test. So it is possible to calculate Fisher’s exact test with more than 2 modalities in one variable.

It is written in Table 1 that Ficher’s exact test was used. I think this is not the case for Diagnosis according the author response.

Results

It is possible to have standardized decimal point in p value.

0.35 Could be witten 0.350.

There is a confusion in the article between total population (for me the 7310 subjects) and the study population (for me the 1258 subjects

Table 1: change the Table title by « Baseline characteristics of study population according to SARS-Cov-2 serology

Table 2: change the Table title by « Baseline characteristics of total population by study participation

7. PLOS authors have the option to publish the peer review history of their article (what does this mean?). If published, this will include your full peer review and any attached files.

Reviewer #1: No

Reviewer #2: No

Reviewer #3: No

Reviewer #4: No

---

## [Author Response · Author response to Decision Letter 1]

2 Feb 2022

We thank all Reviewers for once again carefully revising our manuscript and for raising some important and relevant questions. Please find our specific comments below. Thank you.

6. Review Comments to the Author

Reviewer #1: The word adjusted could still be used as alternative to the word corrected.

Author response: revised accordingly. Thank you for the comment.

For Table 1 & 2, chi-squares test where applicable could still be utilized provided the assumptions of the statistical test are fulfilled/met. Please check the assumptions and also recheck the p values. If Fisher's Exact test is used, one sided or two sided p value to be stated.

Author response: Good point. We have now added the sentence: “P-values for Fisher’s exact tests were two-sided”.

Reviewer #2: Revisions were adequate Revisions were adequate Revisions were adequate Revisions were adequate Revisions were adequate

Reviewer #3: (No Response)

Reviewer #4: I thank the authors for their responses to comments. Nevertheless, I have some other comments.

Introduction

If you want to use the acronym SMI, it is best to associate it with severe mental illness as in the abstract. Otherwise use SMD for severe mental disorders.

Author response: Thank you for your observation. You are completely right. Severe mental disorders have now been changed to severe mental illness.

Methods - Statistical analysis

Due to the small number of expected subjects (<5) in some categories, and the highly unbalanced marginal

sums of the actual data set, Pearson’s chi-square test cannot be applied. It is better to use Fisher’s exact test.

Author response: Fisher’s exact test could not be made with too many groups or observations.

Reviewer comment : The extension of Fisher's exact test to the case where the two variables have any finite number of modalities, but greater than two, was first performed by G. H. Freeman and J.H. Halton in 1951. This test is sometimes referred as the Freeman-Halton or Fisher-Freeman-Halton test. So it is possible to calculate Fisher’s exact test with more than 2 modalities in one variable.

It is written in Table 1 that Ficher’s exact test was used. I think this is not the case for Diagnosis according the author response.

Author response: We thank you for the comment and apologize for the misunderstanding regarding the statistical analyses used in the manuscript. After the first comment wisely suggesting Fisher’s exact test as test when comparing categorical demographics (in table 1 and table 2), the authors applied this test together with an unpaired t-test (for continuous demographics). The authors recognize that this was not clearly stated in our responses to the review comments. As written in table 1 and table 2, and the statistical analyses section, Fisher’s exact tests were used.

Results

It is possible to have standardized decimal point in p value.

0.35 Could be witten 0.350.

Author response: All p-values are now standardized to contain three decimals. 

There is a confusion in the article between total population (for me the 7310 subjects) and the study population (for me the 1258 subjects

Table 1: change the Table title by « Baseline characteristics of study population according to SARS-Cov-2 serology

Table 2: change the Table title by « Baseline characteristics of total population by study participation

Author response: Thank you. This has now been corrected as suggested.

---

## [Editor Report · Decision Letter 2]

9 Feb 2022

SARS-CoV-2 seroprevalence among patients with severe mental illness: a cross-sectional study

PONE-D-21-26611R2

Dear Dr. Sass,

We’re pleased to inform you that your manuscript has been judged scientifically suitable for publication and will be formally accepted for publication once it meets all outstanding technical requirements.

Kind regards,

Ray Borrow, Ph.D., FRCPath

Academic Editor

PLOS ONE
---

## [Editor Report · Acceptance letter]

21 Feb 2022

PONE-D-21-26611R2 

SARS-CoV-2 seroprevalence among patients with severe mental illness: a cross-sectional study 

Dear Dr. Sass:

I'm pleased to inform you that your manuscript has been deemed suitable for publication in PLOS ONE. Congratulations! Your manuscript is now with our production department. 

Kind regards, 

on behalf of

Prof. Ray Borrow 

Academic Editor

PLOS ONE